# Mining and Local Economies: Dilemma between Environmental Protection and Job Opportunities

**Angelo Antoci [1], Paolo Russu [1,*] and Elisa Ticci [2]**

[1] Department of Economics and Business, University of Sassari, via Muroni 25, 07100 Sassari, Italy; antoci@uniss.it

[2] Department of Political and International Sciences, University of Siena, via P.A. Mattioli 10, 53100 Siena, Italy; ticci4@unisi.it

* Correspondence: russu@uniss.it

**Abstract:** Mining areas often experience a climate of social tension due to the potential trade-off between expected employment impact and concerns for environmental damage. We address this topic from a theoretical perspective that, unlike most empirical research, includes medium-term dynamics. We developed a two-sector dynamic model that provides a new way to identify differences among mining regions in terms of conflict risk, local development, and welfare. There are critical points in the natural-resource base of local nonmining activities and in the pollution rate of mining operations, which determine the type of dynamics and its welfare outcomes due to the opening up of the economy to mining investment. Pollution control is a sine qua non for welfare gains despite new job opportunities in the mining sector.

**Keywords:** environmental externalities; mining investments; resource-based regions; local development

---

## 1. Introduction

The global integration of economies and the rising demand for raw materials and commodities, mainly driven by urbanization, population, and economic growth, have increased the exposure of rural economies to investments in mining operations. Since the beginning of the 2000s, global production of many mineral products has substantially increased (see data in [1]), while huge investments have taken place in Latin America, Africa, and parts of Asia, leading to a partial shift in mining locations from the developed to developing and transition countries (International Council on Mining and Metals, ICMM [2]).

The environmental footprint of mining has therefore increased, especially in rural areas of developing countries where local poor communities interact with large investors, even transnational corporations, and where the environmental consequences of mining are a main reason for concern. Mining operations often require the intensive use of water resources, are land-demanding, and can create heavy environmental externalities, including soil erosion and contamination, air and water pollution from acid mine drainage, to leakage of chemicals and sedimentation. Indeed, this growth in global exploration and mining activity has been accompanied by a significant increase in conflicts associated with mining operations [3]. The Environmental Justice Atlas [4] reports, for instance, that out of 2922 cases of globally recorded environmental conflicts, 592 cases were related to mineral ores and building extraction (the Environmental Justice Atlas documents and catalogs social conflict around environmental issues. For details, see [5]). On the other hand, mining companies are increasingly aware of the need to improve their relationship with communities in which they operate [6] and of the

crucial role of sustainability problems in this challenge. Environmental stewardship is a priority for the ICMM. For mining companies, gaining trust, acceptability, and public consent for their operations, namely, obtaining a "Social Licence to Operate" (SLO) is a complex, dialogic, and reflexive process [7], but it is essential to avoid potentially costly conflicts and risks. Consulting company EY classified the SLO as the greatest business risk facing mining and metal companies in 2019–2020 [8], while a comparative case study by Prno [9] on SLO determinants in the mining industry concluded that sustainability is a dominant concern for communities.

Science research on mining has responded with an increasing number of works on the need for a deeper knowledge of socioenvironmental sustainability dynamics linked to this unprecedented rise in mining activity in the rural areas of developing countries. We set ourselves the difficult goal of making a small contribution to this remarkable body of literature along two main directions. First, we applied the tools of optimization economic modeling to understand how environmental factors influence the development paths of mining and other environmentally dependent activities, and their resulting local-welfare outcomes. The bulk of science research on mining consists of empirical- or conceptual-framework [10,11] analysis. This is sensible, as most mining communities experience a complex set of interactions between a number of private and public stakeholders (see, for instance, [12]) that do not adapt very well to model simplification. However, the "simplicity" of an optimization model allows us to logically isolate and sort out different dynamics, which are simultaneously at stake, and to identify their essence while at the same time discussing its development over time. The lack of consistent data covering long periods of time usually undermines this exercise, and this may be a strong limitation in the case of mining, typically perceived as a "dirty" industry, since environmental dynamics are often characterized by cumulative processes and lagged effects. We chose our simplifying assumptions in order to focus on two of the dimensions, which the literature on local effects of mining has identified as relevant: employment and environmental impact. Indeed, a recent literature review on the social impact of the mining sector [13] found that employment is the main benefit detected by empirical research, while land use and environmental damage are the most concerning social aspects.

As a second aspect of novelty in this paper, we jointly considered two main streams of research in this area: the role of mining in the economic and environmental sphere. Karakaya and Nuur [14], for instance, after reviewing 483 papers, concluded that the literature has shifted from the role of the mining sector for economic development (within the debate on industrialization, technological progress, or institutional quality) to new streams of research on socioenvironmental sustainability. In our model, the mining sector plays an economic role not only through its direct effect on labor employment, but also through its environmental impact, which, in turn, affects labor productivity in the nonmining sector. The result is a model in which the revenues of local populations are determined by the combination of these two forces. In most cases, science research on mining investigated health and environmental impact or socioeconomic effects, either directly produced within the mining sector, or through forward and backward linkages with manufacturing/service sectors or through the fiscal channel. In contrast, few studies have estimated the economic impact due to the environmental externalities of mining activities on other sectors, which is the central topic of the present work.

Against this background, more efforts are needed in both analytical and empirical research to understand how the development processes and dynamics of local economies are driven and shaped by environmental pressures exerted by large extractive companies on resource-dependent nonmining activities. This is appropriate since, in several developing countries, a large share of the labor force is engaged in agricultural or resource-based activities (in 2018, in low-income countries, agriculture accounted for 63% of employment [15]) and natural resources are an important source of income for many households [16]. Furthermore, both earlier [17–20] and recent [21–27] development thinking has mostly recognized the central role of agriculture productivity in poverty reduction and economic development.

We devised a two-sector dynamic model showing that environmental externalities can represent a discriminating factor in the determination of the development path followed by a mineral-rich local

economy. We analyzed in detail the most likely trajectory, followed by a local community specialized in the use of renewable resources when, starting from a stationary state of low-level consumption and low-level environmental pressure, it experiences a sudden inflow of investments in mining activities, which harm the environment. We found three possible development patterns and identified critical points in carrying capacity of renewable resources and in the pollution rate of mining that determine which one is selected. Differences in these two factors can give rise to substantially different welfare and environmental outcomes, from a downward spiral of impoverishment and environmental collapse to a transition towards a stationary state with a richer and differentiated economy and positive levels of resources. In particular, we found that, in the case of intermediate levels of renewable natural capital, the low-pollution intensity of mining is a prerequisite for the opening of mines to produce local welfare improvements, regardless of the labor intensity of mining. If this condition is not met, the economic system is likely to follow a path of specialization in the mining sector associated with a reduction in the welfare of the local population.

The rest of the paper is structured as follows. Section 2 provides a brief overview of the related literature. Sections 3–5 present the model and its analysis. Section 6 explains the economic content and implications of the analytical results by discussing the different development and welfare paths that can be generated. Section 7 concludes with policy considerations and suggestion for future research. Analysis of the model is in Appendix A.

## 2. Related Literature

Before presenting our model, this section sets out the broad context in which it is located. The role of large-scale mining in local rural communities of developing countries is a topic of inquiry which is receiving increasing attention in the literature. Chuhan-Pole at al. [11], in a recent literature review, identified three main channels through which the abundance of mineral resources affects local areas. First, the market mechanism refers to the impact on demand for jobs or other local inputs and their multiplier effect on nonmining sectors through backward and forward linkages or other spillovers. Second, the fiscal channel is mainly represented by an increase in fiscal revenues, whose positive or negative effects (for instance, respectively, increased public spending or corruption) are usually influenced by institutional quality. Finally, the environmental channel outlines the impact that pollution and natural resources use during mining activities may have on human or animal health, environmental quality, and endowments of natural resources or access to them.

Theoretical studies modeling these effects are very limited and focus on the market mechanism. Some works (see, e.g., [28–30]) investigate the design of optimal local content policies in the context of extractive industries. Mainly via the Nash bargaining theory, these works analyse games in which two players—the host government and the mining company—negotiate the terms of concession contracts. A Nash bargaining solution is reached when a contract is designed so that no one has incentive to break. They aim to analyze the links between optimal local content policies and several elements characterizing the host economy, for example, the productivity of local suppliers. Ghebrihiwet and Motchenkova [31] modeled technology transfers from foreign multinationals to local firms under different market structures and foreign direct-investment policies, highlighting the policies that could improve the host country's welfare. Di Corato [32] proposed a Nash bargaining model, allowing one to characterize optimal investment choices of a foreign multinational facing the threat of nationalization of the extractive industry. Our paper may be collocated in this area of research, that is, among the theoretical works dealing with local spillover effects of extractive industries on host economies. However, to the best of our knowledge, our model is the only one that analyzes causal links among activities of extractive industries, environmental degradation generated by them, and performance of local economic activities depending on a (renewable) natural resource.

Empirical research has made greater progress in the study of the three channels, but there is still less of an advance in our understanding of how the environmental and channels interact with each other. Indeed, many studies have investigated a large spectrum of the local socioeconomic impact of the

mining sector: local economic growth, agglomeration economies, demand shock, and multiplier effects on other sectors [11,33–36], local output and employment composition ([11,37]), local corruption and violence ([38–41]), living standards, poverty, and inequality, [11,35,42–44], and local public spending and services [45]. At the same time, a literature stream on the impact of mining on the local environment is emerging. Important effects of mining on land use and contamination, for instance, have been found in some mining regions in India [46], Chile [47], and China [48,49], while other studies have documented a significant negative impact on human health outcomes and risks. Ouoba [50] showed that approximately 40% of the gold-mining industry's added value in Burkina Faso represents natural capital depreciation, which is due to both the depreciation of the gold stock and the health cost of pollution. Der Goltz and Barnwal [51], analyzing microdata from about 800 mines in 44 developing countries, found evidence that mining communities exhibit an increased incidence of health problems, which are known consequences of exposure to heavy metals, anemia among women, and stunting in young children.

A less investigated area is the potential role of the environmental damage of mines in producing economic externalities on other sectors that, in turn, affect the structure of local economies and the economic welfare of local populations (see also [52] on this point). Only few empirical studies, for instance, have estimated the impact of mining on the productivity of agricultural and other resource-based economic activities. Ticci and Escobal [42] found that mining companies in Peru have nonstatistically significant impact on agricultural production and prices. Similarly, Andersson et al. [53] concluded that in Ghana, Mali, Tanzania, and Burkina Faso, a greenness index, which is correlated with agricultural production, does not decrease within the proximity of large-scale gold mines. In contrast, Aragón and Rud [54] estimated that gold mining in Ghana reduces the productivity of nearby farming activities due to environmental damage. A negative impact of coal mining on agricultural productivity has also been found by Mishra and Pujari [55] in the Indian state of Orissa. Using subjective data, analysis by Li et al. [56] of host communities in Shanxi, China revealed that coal mining has a negative impact on a wide range of wellbeing factors pertaining to the natural environment and the economy. Metero [57] showed that mining-related land dispossessions in Mapela, Limpopo, South Africa, disrupted the local livelihood system, limiting the ability of rural households to accumulate, especially from agriculture. In conclusion, evidence of the role of mining pollution on surrounding resource-dependent activities, primarily the productivity of local agriculture, is still scant and inconclusive, even though it has given significant warnings. We complement this stream of the literature by using theoretical tools that are less constrained by data shortage and measurement difficulties in order to explore the economic scope of the environmental channel and to jointly consider environmental and market dynamics. Among market factors, we focus on direct employment generated by mining activities, while all remaining market and fiscal linkages, extensively investigated by the earlier empirical literature, are excluded for reasons of analytical tractability.

## 3. Model Setup

We propose a model of an economy with two sectors, the mining sector and the local sector. Prices of goods produced by the two sectors are normalized to 1 and considered as exogenously given by agents. The mining sector extracts mineral resources using physical capital and wage labor, while the local sector is represented by traditional activities that rely on self-employed labor, land, and renewable natural resources. To fix ideas, we define the local sector as the farming sector, which typically depends not only on land, but also on renewable natural resources and environmental regulating services (water, genetic diversity, plants' regeneration capacity, water regulation, watershed resilience, biodiversity, soil protection and nutrient circulation, pest control, and pollination). There is no population growth, while agents belong to two different communities: external investors (I-agents) (it is worth emphasizing that, in this model, we use the term "external" to refer not just to foreign investors, but also to national entrepreneurs whose capital is derived from a source outside the local economy) and local agents (L-agents). Both communities consist of identical individuals. For simplicity, we assume that I-agents

cannot invest in the local sector. Under the standard neoclassical assumption of competitive capital markets [58], investors can rent capital in spot markets and take the rental price of capital as given. I-agents also hire the labor force provided by L-agents. L-agents are endowed only with their own working capacity, which they use partly to work as employees for external investors and partly as producers in the local sector.

The production functions of the two sectors are increasing at decreasing rates in their inputs. By normalizing the fixed amount of land endowment to 1, the production function of the representative L-agent is given by:

$$Y_L = E^\beta L^{1-\beta},$$

where,

　　$E$ is the stock of the free-access renewable environmental resource;

　　$L$ is the amount of time the representative L-agent spends on local-sector production; and

　　$0 < \beta < 1$ holds.

The L-agent's total endowment of time is normalized to 1, and leisure is excluded; thus, $1 - L$ represents the L-agent's labor employed by the representative I-agent as wage labor. The production function of the representative I-agent is represented by constant returns to the scale function as follows:

$$Y_I = \delta K_I^\gamma (1 - L)^{1-\gamma}, \tag{1}$$

where $K_I$ denotes the stock of physical capital invested by the representative I-agent in the economy, $\delta > 0$ is a productivity parameter, and $1 > \gamma > 0$. $\delta$ is also a parameter representing local-mineral-resource wealth. The role of mineral resources is embodied in the coefficient $\delta$. Mineral-resource abundance implies a higher $\delta$ that, in turn, means higher productivity in the mining sector. We focus on the medium term, during which mining extracts only a negligible part of the mineral-resource stock. In other words, in the considered time horizon, the mining sector does not experience increasing costs or constraints due to mineral-resource exhaustion, so $\delta$ is fixed.

We assume that the environmental impact of the local sector is nil since our focus is on those scenarios where the environmental damage of the local agents' production is unimportant when compared to that of capital-intensive mining activities. This is not to deny that all economic activities, including subsistence farming, have environmental impact. However, the view of poverty as a major cause of environmental degradation is outdated [16,59]. In addition, in many cases, poor farmers in developing countries rely on a low-to-null use of synthetic chemical pesticides or fertilizers, while adopting techniques that, although noncertified, are de facto organic or near-organic agriculture [60]. The evolution of renewable environmental resources is governed by a standard logistic function (see [61]), modified by taking account of the impact of the mining sector:

$$\dot{E} = E(\overline{E} - E) - \eta \overline{Y}_I \ \text{ if } \ E > 0 \tag{2}$$

$$\dot{E} = 0 \ \text{ if } \ E = 0$$

where parameter $\overline{E}$ represents the long-run level that environmental resource $E$ would approach in the absence of the negative effect generated by the mining sector. That is, $\overline{E}$ measures the carrying capacity of the renewable resource.

Each economic agent considers the effect of their choices on the dynamics of $E$ to be negligible and do not take it into account. In other words, each agent sees the evolution of $E$ as exogenously determined when solving their maximization problem. In addition, working under the assumption that each population of agents consists of identical individuals, value of average output $\overline{Y}_I$ coincides (ex post) with per capita value $Y_I$.

This is a stylized scenario, but it represents the main differences between the sets of options that local populations and external investors can use to generate their income flows. Large mining companies managed by external investors are usually capital-intensive activities (often transnational

companies) that are able to gain access to capital markets. In contrast, the use of labor-intensive techniques, employment of family labor, and constraints in access to credit markets are typically key features of the production activities of local communities. Local populations have few defensive strategies and are vulnerable to environmental degradation. Barbier [62], for example, summarizes his review of the empirical literature on the relationship between poverty and natural resources in developing countries by observing that the rural poor are almost "assetless", they depend "critically on the use of common-property and open-access resources for their income, they rely on small plots of lands, and on selling their labor, which is their only other asset". We modeled and simplified these settings by assuming (see [63]) that local agents cannot accumulate physical capital rather, they can only rely on two productive inputs, namely, their labor and natural capital (renewable environmental resource $E$ and land, which was normalized to 1). Local agents can react to a reduction in labor productivity in the local sector due to a depletion of $E$ by substituting self-employed labor in the local sector with wage labor. However, this strategy is of limited effectiveness because it cannot be indefinitely adopted (the total amount of labor that can be employed by each local agent is fixed). In addition, it can have negative side effects due to the degradation of the environment.

## 4. Choices of Agents and Dynamics

The representative L-agent, in each instant of time $t$, has to choose the value of $L$ in order to maximize the value of the objective function:

$$\Pi_L = E^\beta L^{1-\beta} + (1-L)w, \tag{3}$$

where $w$ is the wage rate.

The representative I-agent chooses their labor demand $1 - L$ and stock of physical capital $K_I$ that they invests in the economy in order to maximize the profit function:

$$\Pi_I = \delta K_I^\gamma (1-L)^{1-\gamma} - w(1-L) - rK_I,$$

where parameter $r > 0$ represents the cost of $K_I$.

Both $w$ and $r$ are considered as given by the agents. However, wage $w$ is endogenously set in the economy by the labor-market equilibrium condition (we excluded the import of labor from other economies), while $r > 0$ is an exogenous parameter.

The above-described choice problems are analyzed in Appendix A.1. According to this analysis, market clearing choice $L^*$ of $L$ is:

$$L^* = \min\{1, \Gamma E\},$$

where:

$$\Gamma := \left[ \frac{1-\beta}{\delta(1-\gamma)\left(\frac{\gamma\delta}{r}\right)^{\frac{\gamma}{1-\gamma}}} \right]^{\frac{1}{\beta}} \tag{4}$$

Note that $L^* > 0$ if $E > 0$; this excludes the specialization of the economy in mining if $E > 0$. In the $E > 0$ context, two cases can be distinguished: the case without specialization in the local sector (that occurs when stock $E$ is low enough, that is, $\Gamma E < 1$, and, consequently, $1 > L^* > 0$ holds) and the case with specialization in the local sector (that occurs when $\Gamma E \geq 1$, and, consequently, $L^* = 1$ holds).

Investment and output levels chosen by the representative I-agent are, respectively, given by:

$$K_I^* = \left(\frac{\gamma\delta}{r}\right)^{\frac{1}{1-\gamma}}(1-L^*)$$

$$Y_I^* = \delta K_I^{*\gamma}(1 - L^*)^{1-\gamma} = \delta \left(\frac{\gamma\delta}{r}\right)^{\frac{\gamma}{1-\gamma}} (1 - \Gamma E)$$

By substituting $Y_I^*$ for average product $\overline{Y}_I$ in Equation (2), we found that, in the context of $\Gamma E < 1$, Equation (2) can be written as follows:

$$\dot{E} = E(\overline{E} - E) - \eta\delta \left(\frac{\gamma\delta}{r}\right)^{\frac{\gamma}{1-\gamma}} (1 - \Gamma E) \tag{5}$$

while, in the context of $E \geq 1/\Gamma$ (as the representative L-agent spends all their time endowment working in the local sector, $L^* = 1$), Equation (2) becomes:

$$\dot{E} = E(\overline{E} - E) \tag{6}$$

## 5. Dynamic Regimes

According to Equation (5), $\dot{E} = 0$ holds if $f(E) = g(E)$, where:

$$f(E) := E(\overline{E} - E)$$

$$g(E) := \eta a\, (1 - \Gamma E)$$

$$a := \delta \left(\frac{\gamma\delta}{r}\right)^{\frac{\gamma}{1-\gamma}} \tag{7}$$

Graphs of $f(E)$ and $g(E)$ can have, at most, two intersection points, and therefore at most two stationary states in which $E > 0$ exists. Figures 1 and 2 illustrate all the possible dynamic regimes that can be observed under Dynamic (5) and (6) (see Appendix A.2). Threshold values of parameters $\overline{E}$ and $\eta$ to which Figures 1 and 2 refer are:

$$\frac{1}{\Gamma} = \left[\frac{\delta(1 - \gamma)\left(\frac{\gamma\delta}{r}\right)^{\frac{\gamma}{1-\gamma}}}{1 - \beta}\right]^{\frac{1}{\beta}} \tag{8}$$

$$\overline{E}_T := 2\sqrt{a\eta} - a\eta\Gamma \tag{9}$$

$$\eta_0 := \frac{1}{a\Gamma^2} \tag{10}$$

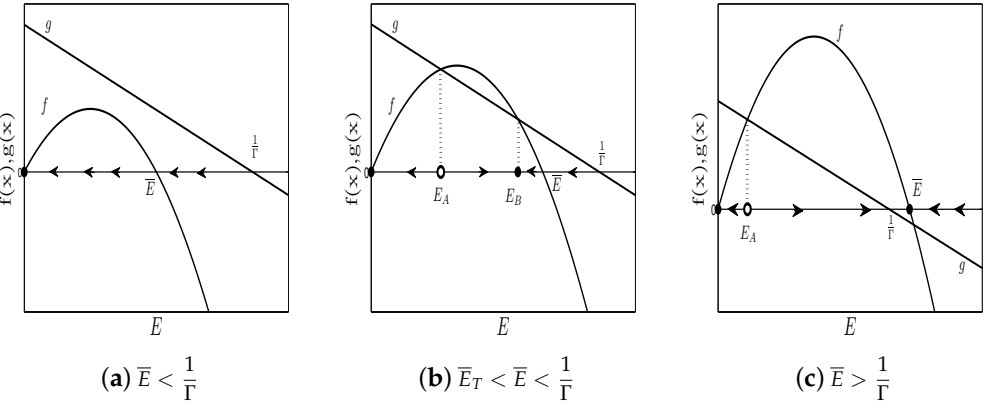

(a) $\overline{E} < \frac{1}{\Gamma}$  (b) $\overline{E}_T < \overline{E} < \frac{1}{\Gamma}$  (c) $\overline{E} > \frac{1}{\Gamma}$

**Figure 1.** Dynamic regimes in the context $\eta < \eta_0$, obtained by varying the parameter $\overline{E}$.

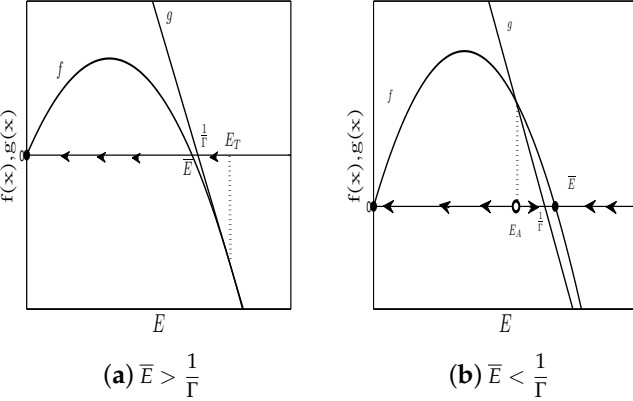

**(a)** $\overline{E} > \dfrac{1}{\Gamma}$              **(b)** $\overline{E} < \dfrac{1}{\Gamma}$

**Figure 2.** Dynamic regimes in the context $\eta \geq \eta_0$, obtained by varying the parameter $\overline{E}$.

Figures 1 and 2 illustrate the regimes that can be observed if the environmental impact of the mining sector (measured by parameter $\eta$) is, respectively, lower and higher than the threshold value $\eta_0$ (see Equation (10)). The taxonomy of dynamic regimes is based on the value of parameter $\overline{E}$, representing the carrying capacity of the environmental resource.

In the figures, full dots (●) and empty dots (○) represent, respectively, attractive and repulsive stationary states. Throughout the paper, we focus our analysis on the dynamics of an economy starting from initial position $E(0) = \overline{E}$, where $\overline{E}$ is the carrying capacity. This allows us to interpret part of the results as the effects of opening up the economy to inward flows of capital for mining investment. Observe that the following dynamic regimes can occur:

(a)    Dynamic regime illustrated in Figures 1 and 2, where stationary state $E = 0$ is globally attractive. In this regime, the trajectory starting from $E(0) = \overline{E}$ approaches $E = 0$. Thus, opening the economy to mining operations leads to the complete exhaustion of renewable natural resources and the crowding out of the local sector. This dynamic regime is observed when the carrying capacity (measured by parameter $\overline{E}$) of the natural resource is very low ($\overline{E} < \overline{E}_T < 1/\Gamma$ if $\eta < \eta_0$, $\overline{E} < 1/\Gamma$ if $\eta \geq \eta_0$);

(b)    Dynamic regimes illustrated in Figures 1b,c and 2b, characterized by the coexistence of two locally attractive stationary states (bistable regimes): stationary state $E = 0$ and either stationary state $E = \overline{E}$ or stationary state $E_B < \overline{E}$. The basins of attraction of the two attractive stationary states are separated by repulsive stationary state $E_A$. In stationary state $E_B$, both sectors coexist, while in stationary state $E = \overline{E}$, the economy specializes in the local sector. Bistable regimes take place only if carrying capacity $\overline{E}$ of the natural resource is high enough. In this context, the initial condition $E(0)$ plays a key role in determining the time evolution of $E$. The economy, starting from $E(0) = \overline{E}$, either remains in $\overline{E}$ or approaches $E_B$. The former scenario occurs in economies with a very high carrying capacity ($\overline{E} > 1/\Gamma$, Figure 1c), or high pollution impact of the mining sector ($\eta > \eta_0$, Figure 2b) which, after the opening up the economy to external capital, remains specialized in the local sector. The latter scenario, namely, the transition towards a diversified economy in which both sectors coexist, is observed in economies with an intermediate level of carrying capacity ($\overline{E}_T < \overline{E} < 1/\Gamma$) and exposed to low-pollution-intensity capital inflows in the mining sector ($\eta < \eta_0$).

In a bistable regime context, the economy approaches stationary state $E = 0$ only if the initial value $E(0)$ belongs to interval $[0, E_A)$. This may be the case when an exogenous environmental shock lets the economy, although initially not open to external investments, start from an initial value $E(0)$ lower than $E_A$ (and, consequently, lower than carrying capacity $\overline{E}$). If this occurs, the low value of $E(0)$ gives rise to a self-reinforcing growth process of labor input in the mining sector and investment $K_I$, which drives the economy towards the complete depletion of the environmental resource.

As far as state variable $K_I$ is concerned, note that equilibrium value $K_I^*$ of $K_I$ is determined by equation $K_I^* = \left(\frac{\gamma\delta}{r}\right)^{\frac{1}{1-\gamma}} (1 - L^*)$. Thus, $K_I^*$ is negatively related to $L^* = \min\{1, \Gamma E\}$ and, consequently, to $E$. The stationary state with $E = 0$ is, therefore, associated with the highest value of $K_I^*$. Figure 3 shows the dynamics in the plane $(E, K_I)$ corresponding to the bistable regime illustrated in Figure 1b.

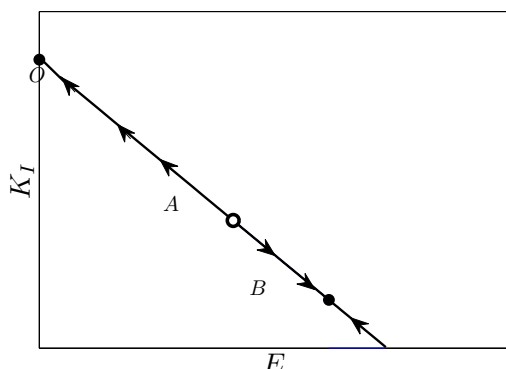

**Figure 3.** Dynamics in plane $(E, K_I)$ corresponding to bistable regime illustrated in Figure 1b.

## 6. Welfare Implications: Root of the Dilemma

Revenues of L-agents are composed of two addends: labor remuneration in the mining sector and local-sector output. Capital inflows expand labor demand in the mining sector, exerting pull forces on labor and up-pressure on wage rates. At the same time, growth in mining operations causes environmental degradation, in turn affecting the productivity of the local sector. This effect boosts the labor supply to the mining sector and depresses wage rates. The net impact on revenues of L-agents is, therefore, not univocal and it might not be directly connected with environmental dynamics despite the reliance of the local producers on natural resources. This section clarifies which and under which conditions different local welfare dynamics are generated by the model. We focus on the welfare consequences of openness to mining investments (in our analysis, we concentrated on the revenues of L-agents, since the remuneration of each unity of external investment is exogenously determined: $\gamma Y_I / K_I = r$) and start by comparing the level of L-agent's revenues in reachable stationary states.

### 6.1. Local Welfare, Environmental Outcomes, and Physical Capital Stock in Attractive Stationary States

According to results illustrated in Appendix A.1, we find that, in the context in which the economy does not specialize in the local sector (i.e., when $\Gamma E < 1$ or, equivalently, $E < 1/\Gamma$), the equilibrium wage rate is constant and given by $w = a(1 - \gamma)$, where $a := \delta \left(\frac{\gamma\delta}{r}\right)^{\frac{\gamma}{1-\gamma}}$. Furthermore, revenues $\Pi_L(E)$ of L-agents are $\Pi_L(E) = E^{\beta}$ when $E \geq 1/\Gamma$ and $\Pi_L(E) = \left[\Gamma^{-\beta} - a(1 - \gamma)\right] \Gamma E + a(1 - \gamma)$ when $E < 1/\Gamma$, where $\left[\Gamma^{-\beta} - a(1 - \gamma)\right] > 0$. That is, revenues $\Pi_L(E)$ of the representative L-agent are strictly increasing in $E$ for every $E \geq 0$. Thus, in attractive stationary state $E = 0$, revenues are lower than in other attractive stationary states (either $E_B$ or $E = \bar{E}$) in bistable regimes as illustrated in Figure 1b,c and 2b (see Proposition A2 in Appendix A.3).

The equilibrium value of $K_I$ is determined by equation $K_I^* = \left(\frac{\gamma\delta}{r}\right)^{\frac{1}{1-\gamma}} (1 - L^*)$, thus $K_I^*$ is negatively related to $L^* = \min\{1, \Gamma E\}$ and, consequently, to $E$. Furthermore, according to the results stated above, an increase in $K_I$ is always associated with a reduction in $E$ and $\Pi_L$. This means that environmental degradation caused by the mining sector had a negative impact on L-agents' revenues. This negative impact is always larger than the positive impact of the rise in labor productivity caused by an increase in $K_I$. As a result, in bistable regime contexts (illustrated in Figures 1b,c and 2b), in locally attractive stationary state $E = 0$, workers' revenues reach the lowest possible level, while external investment $K_I$ reaches the highest possible value (see Figure 3). Along the trajectory approaching

$E = 0$, the effort of L-agents to defend themselves from environmental degradation drives the economy towards complete specialization in the mining sector.

*6.2. Openness to Mining Investments: Different Environmental Contexts, Different Welfare Outcomes*

The inverse relation between $K_I$ and $\Pi_L$, as detected in the previous section, does not mean that full specialization in the local sector always produces the highest level of welfare for L-agents. The effects on L-agent revenues, generated by external mining investments, can be better understood by comparing the generated dynamics by the model, and the dynamics according to which $L = 1$ and $K_I = 0$ hold for every $E \geq 0$, that is, the dynamics that would be observed in the absence of extractive operations, financed by external investors. Under these dynamics, revenues of L-agents are given by $E^\beta$, while $E = \overline{E}$ is always a globally attractive stationary state (if the environment is not affected by the polluting activity of mining, the time evolution of $E$ is described by the logistic Equation (6)). In the absence of mining, therefore, the local economy always converges to a stationary state in which revenues of L-agents are equal to $\overline{E}^\beta$. Figure 4 aims to clarify how L-agents may be better off with mining investments rather than by being fully specialized in the local sector, even if L-agents' revenues are increasing in $E$. In this figure, revenues of L-agents in the absence of mining investments ($E^\beta$) and those with them ($\Pi_L(E)$) are represented as functions of variable $E$. If the economy converges toward stationary state $E'_B$, then it holds that $\Pi_L(E'_B) > \overline{E}^\beta$; in other words, L-agents reach a higher level of welfare in the case of openness to mining investments than in a closed economy without inflows of external capital. On the contrary, if it converges towards $E''_B$, with $E''_B < E'_B$, it holds $\Pi_L(E''_B) < \overline{E}^\beta$. However, under which conditions does openness to extractive activities improve the livelihoods of local agents?

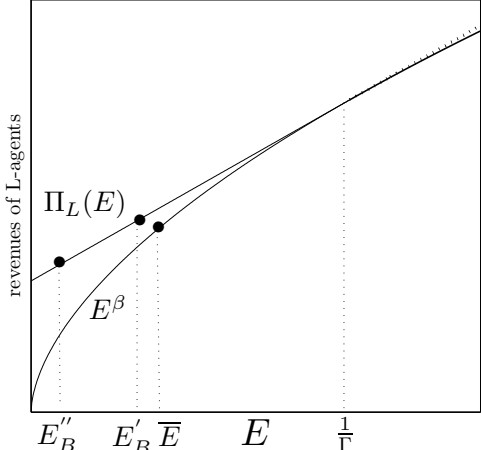

**Figure 4.** Revenues of L-agents in the absence of external investments ($\Pi_L^{ND}(E) = E^\beta$) and those with external investments ($\Pi_L(E)$), represented as functions of variable $E$.

Analysis illustrated in Appendix A.3 answers this question, as it shows that opening up to external capital has differentiated welfare and environmental effects according to the level of carrying capacity (measured by parameter $\overline{E}$) and the pollution intensity of the mining sector (measured by parameter $\eta$).

The following taxonomy of scenarios clarifies this statement. In this case, we focus, as in the previous section, on the trajectory followed after opening the economy to inward flows of capital for mining investment, that is, in an economy starting from initial position $E(0) = \overline{E}$.

*Case 1: Very high level of environmental carrying capacity*

If $\overline{E} > 1/\Gamma$ holds (Figures 1c and 2b), then the economy starting from $E(0) = \overline{E}$ remains in $\overline{E}$, where $K_I = 0$. L-agents are not willing to provide their labor force as workers at acceptable wages for the I-agents. They would renounce their traditional activities only in exchange for very high compensation. Consequently, mining investments had no effect on the environment and the welfare of local agents. Reality is not so simple: mining companies may invest anyway by importing labor from other regions (a possibility that in our model is excluded by construction). This case could manifest itself as strong (successful or otherwise) opposition to mining operations. Rather than the absence of mining investments, this analytical case may in effect mirror the fact that indigenous or local communities living in rainforests or in areas very suitable for cultivation are often among the most strenuous opponents to mining activities.

For instance, in the Peruvian Amazon between 2009 and 2012, the share of indigenous territorial reserves covered by hydrocarbon concessions was largely reduced, suggesting that companies might be giving up those concessions in which indigenous groups have territorial claims. In contrast, overlaps between concessions and watersheds have increased over time in water-constrained regions of Peru [64]. Meanwhile, a report on 24 case studies of mining conflicts from 18 different countries [65] found that, in water-rich areas where benefits of ecosystem services are of great importance to local populations, the level of conflict tends to be high, even when the mining project is not fully operational and its impact is likely but not yet realized. This suggests that: (1) the main objective of the protests is to prevent environmental impact and (2) before entering into a trade-off between employment opportunities in mining-related activities and livelihoods dependent on the biophysical environment, local populations prefer, ex ante, this latter possibility.

*Case 2: Very low level of environmental carrying capacity*

This case refers to dynamic regimes illustrated in Figures 1a and 2a. Under these regimes, the trajectory starting from $E(0) = \overline{E}$ approaches a stationary state with $E = 0$, where the stock of mining investments $K_I$ reaches its maximum value and local activities were completely crowded out. Furthermore, in stationary state $E = 0$, local welfare is higher than in $E = \overline{E}$ if the carrying capacity of the environmental resource is low enough, that is, if $\overline{E} < [a(1-\gamma)]^{\frac{1}{\beta}}$ ( condition $\overline{E} < [a(1-\gamma)]^{\frac{1}{\beta}}$ implies $\overline{E} < 1/\Gamma$, since $[a(1-\gamma)]^{\frac{1}{\beta}} < 1/\Gamma$). Otherwise, the opposite holds (see Proposition A3 in Appendix A.3) .

This scenario refers to economies characterized by extremely adverse environmental conditions, where opening up to mining investments that allows for inflows of artificial assets is beneficial regardless of their rate of pollution intensity. Although freely accessible and not negatively affected by local producers, environmental resources are so scarce that their complete substitution with artificial inputs is associated with an improvement in local welfare. It is worth observing, however, that this transition also implies a shift from full dependence on natural capital $E$ to full reliance on exogenous factors (following openness to mining investments) such as $r$ (opportunity cost of physical capital investment) and $\delta$ (total factor productivity of mining sector that also captures local mineral resource wealth). In this case, revenues of L-agents in fact consist only of the wage rate, which depends on the exogenous parameters $r$, $\gamma$, and $\delta$ (see Appendix A.1). This case may be portrayed by the example of the region of Antofagasta, Northern Chile, in the Atacama Desert, which is rich in minerals and poor in renewable natural resources. The mining sector, largely financed by foreign companies, has weak productive backward and forward linkages to local firms [66] and low knowledge spillovers [67], as well as putting pressure on water resources [68], while problems with contamination from heavy metals [69] have led to environmental externalities on economic activities of local and indigenous communities [70] and negatively impacted population health [68,71,72]. Despite these problems, at the end of a mining boom period in the 1990s and 2000s, the region was characterized by the highest per

capita income, the lowest Gini coefficient, and the second lowest poverty level in Chile [73]. This result is consistent with a nationwide study [74] that estimated that the mining boom produced a significant positive effect on wages, employment, and poverty reduction in municipalities more exposed to growth in mining. These findings suggest that, in contexts of scarcity of renewable natural resources, notwithstanding environmental externalities, the mining sector is likely to bring about net economic gains.

*Case 3: Intermediate level of environmental carrying capacity*

The remaining setting is characterized by an intermediate level of carrying capacity (see Figure 1b): $\overline{E}_T < \overline{E} < 1/\Gamma$). In this context, an economy starting from $E(0) = \overline{E}$ converges towards stationary state $E_B$, where the two sectors coexist. Convergence towards $E_B$ ensures growth in local agents' revenues only if the environmental impact of mining $\eta$ is sufficiently below the threshold value $\eta_0$ (see Proposition A3 in Appendix A.3). Under these conditions, the economy undertakes a transition to a diversified economic structure that reduced the vulnerability of local agents towards exogenous environmental shocks while at the same time improves their revenues. For higher values of $\eta$, convergence to stationary state $E_B$ gives rise to a reduction in local agents' revenues. Figure 5 provides a graphical illustration of the critical role of the rate of environmental pressure from the mining sector in determining the divide between local gains and local losses. More precisely, Figure 5 shows the revenues of L-agents in the absence of mining investments, evaluated at state $\overline{E}$ ($\overline{E}^\beta$) (in the absence of mining activity, $E = \overline{E}$ is always a globally attractive stationary state), and the value of $\Pi_L(E)$ evaluated at stationary state $E_B$ (which is reached by the trajectory starting from $E(0) = \overline{E}$). We can easily see that $\Pi_L(E_B) > \overline{E}^\beta$ holds for low enough values of $\eta$ while the opposite holds for higher values of $\eta$.

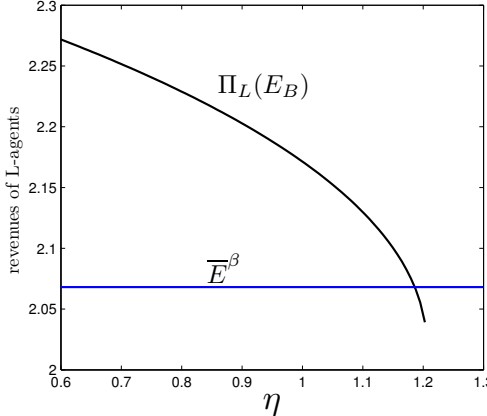

**Figure 5.** Revenues of L-agents in the absence of external investments evaluated at stationary state $E = \overline{E}$ ($\Pi_L^{ND}(\overline{E}) = \overline{E}^\beta$) and those with external investments evaluated at stationary state $E = E_B$ ($\Pi_L(E_B)$), obtained by varying parameter $\eta$. Other parameters fixed at values: $\beta = 0.58$, $\gamma = 0.49$, $\delta = 1$, $r = 0.15$, $\overline{E} = 3.5$.

An empirical example of opening to mining investment in cases with intermediate level of $\overline{E}$ and insufficiently low $\eta$ can be represented by the development of gold mining in Ghana. Ghana indeed reflects the main assumptions of the model and of this scenario: Growing large-scale extractive activities that operate in rural areas where typically traditional farming is the main source of livelihood with little backward and forward linkages with local economies, important environmental pressures from the releases of pollutants in rivers, soil, and air, to pressure on water resources, deforestation, loss of farmland, and even land dispossession [52,75–79]. Ghana has a long history as a gold producer, but the sector has experienced substantial growth, attracting foreign investors, since the early 1990s

until the recent gold rush following the 2008 financial crisis. Using data from household surveys and interviews to key informants, Hausermann et al. [52] found that, in mining communities located along the Offin River, cocoa and subsistence crops, forest, and other land uses were cleared prior to extraction with devastating effects on farming, reduced local food availability, and increased malnutrition. These results are consistent with estimates based on national household surveys by Aragón and Rud [54]. The authors found that the vicinity to mining activity of medium–large enterprises is associated with an increase in rural poverty and a large decrease in agricultural productivity (about 40% between 1997 and 2005). Interestingly, among alternative explanations of the drop in agricultural total factor productivity, they identified mining-related environmental impact as the most plausible. In conclusion, as predicted by our model, social–economic and environmental dynamics in Ghanaian mining areas experienced a process of mining growth that is associated with a welfare-reducing crowding out of traditional rural activities.

## 7. Conclusions

In developing countries, several rural communities could still rely on uncontaminated and preserved environments. The sudden arrival of external investments could provide a large push to escape from poverty, but the positive effect may be temporary if new activities cause environmental shock. This process could be particularly relevant in the case of mining investment. To explore under which conditions that net economic gains or losses are generated, we modeled a simplified local economy with two channels of interaction between resource-based local activities and externally financed mining operations, that is, connections made through environmental dynamics and through the labor market.

The empirical literature has very recently started to investigate the economic role of mining's environmental pressures on rural economic activities in developing countries, with mixed findings. With this model, we found robust theoretical reasons for directing more attention to this aspect and we provided the first taxonomy and conceptual framework to explain this role in different background contexts. We found that environmental factors, either environmental externalities or the abundance of nonmineral renewable resources, could be crucial in shaping the effects of mining investment. In addition, this even holds for an economy in which local populations are able to fully exploit job opportunities created by inflows of mining investments. Local communities in areas with extreme scarcity of renewable resources are most likely to benefit from mining investments (Case 2). Their main risks are to follow a development path that is fully dependent on exogenous factors. In all other cases, analytical results can be interpreted as a warning sign that environmental damage caused by mining should be acknowledged as a key determinant, not only on human health and the ecology of local communities, but also in terms of their economic welfare. If the results from Case 2, with very high environmental carrying capacity, can be read as implicit signs of strong opposition to mining operations, Case 3 provides more clear-cut messages. Under conditions of intermediate levels of environmental carrying capacity, the opening of mines in an area specialized in primary subsistence activities triggers a welfare-improving transition towards a diversified local economy only if the pollution intensity of new mining operations is sufficiently low. For higher rates of environmental impact, the local economy risks to initiate structural change toward specialization in the mining sector, which is associated with a reduction in the welfare of the local population and environmental degradation, even when the local primary sector is not completely crowded out.

Our model supports a clear policy message: limiting the intensity of mining pollution is a prerequisite for welfare improving and non-conflicting coexistence of extractive activities and local resource-based activities, even under ideal conditions, that is, in an economy without barriers to labor mobility towards the mining sector or in the case of renewable-resource abundance. These results, however, need some qualifications that also indicate directions for future research. First, the model does not include the role of fiscal channels, namely, a possible change in public expenditure on local services financed by mining-tax revenues. Thus, it would be interesting to derive the optimal mining

tax in this model under different levels of institutional quality. A classification of the possible scenarios would be modified and results would provide direct policy implications on the mix of mining tax and pollution control. Second, the model considers local populations as a homogeneous block but, in reality, community stakeholders are often more fragmented, with differences between the interests and perspectives of farmers, mining workers, contractors, and local leaders. Finally, with some adaptation, the model may be extended to analyze an emerging issue, namely, livelihood linkages between artisanal mining and farming, especially in rural African areas where, despite the economic centrality of agriculture, engagement in artisanal and small-scale mining is an off-farm diversification strategy [80]. In this case, both local economic welfare would be represented by the sum of miners' and farmers' revenues.

**Author Contributions:** Conceptualization, A.A.; methodology, A.A., P.R., and E.T.; validation, A.A., P.R., and E.T.; formal analysis, P.R.; resources, E.T.; data curation, E.T.; writing—original-draft preparation, E.T.; writing—review and editing, A.A., P.R., and E.T.; supervision, A.A.

**Funding:** This research was supported by the Italian Ministry of Education (MIUR): Dipartimenti di Eccellenza Program (2018–2022), Department of Economics and Business, University of Sassari.

**Conflicts of Interest:** The authors declare no conflict of interest.

## Appendix A

*Appendix A.1. Choice Problem of Agents and Labor-Market Equilibrium*

The representative L-agent in each instant of time $t$ has to choose the value of $L$ in order to maximize the value of the objective function:

$$\Pi_L = E^\beta L^{1-\beta} + (1-L)w$$

Representative I-agent chooses their labor demand $1-L$ and stock of physical capital $K_I$ in order to maximize the profit function:

$$\Pi_I = \delta K_I^\gamma (1-L)^{1-\gamma} - w(1-L) - rK_I$$

An internal solution of the maximization problem of the representative L-agent must satisfy first-order condition:

$$\frac{\partial \Pi_L}{\partial L} = (1-\beta) E^\beta L^{-\beta} - w = 0, \tag{A1}$$

which determines the labor offered by the representative L-agent as a function of $E$ and $w$.

The optimization problem of the representative I-agent gives rise to the following first-order conditions:

$$\frac{\partial \Pi_I}{\partial (1-L)} = \delta(1-\gamma)K_I^\gamma(1-L)^{-\gamma} - w = 0 \tag{A2}$$

$$\frac{\partial \Pi_I}{\partial K_I} = \delta\gamma K_I^{\gamma-1}(1-L)^{1-\gamma} - r = 0 \tag{A3}$$

The labor-market equilibrium condition is therefore given by:

$$\delta(1-\gamma)K_I^\gamma(1-L)^{-\gamma} = (1-\beta) E^\beta L^{-\beta} \tag{A4}$$

By Equation (A3), it holds:

$$K_I = \left(\frac{\gamma\delta}{r}\right)^{\frac{1}{1-\gamma}} (1-L) \tag{A5}$$

and substituting $K_I$ in Equation (A4) we obtain:

$$L = \Gamma E \tag{A6}$$

where:

$$\Gamma := \left[ \frac{1-\beta}{\delta(1-\gamma)\left(\frac{\gamma\delta}{r}\right)^{\frac{\gamma}{1-\gamma}}} \right]^{\frac{1}{\beta}} \tag{A7}$$

Function (A6) identifies labor-market equilibrium value $L^*$ of $L$ if the right side of Function (A6) is lower than 1; otherwise $L^* = 1$, that is:

$$L^* = \min\{1, \Gamma E\}$$

In the context $\Gamma E < 1$, the equilibrium wage rate is constant and given by $w = \delta(1-\gamma)\left(\frac{\gamma\delta}{r}\right)^{\frac{\gamma}{1-\gamma}}$. Indeed, by substituting Function (A6) in Equation (A1) we obtain:

$$\begin{aligned} w &= (1-\beta)\,E^\beta L^{-\beta} \\ &= (1-\beta)\,E^\beta\,[\Gamma E]^{-\beta} \\ &= (1-\beta)\,\Gamma^{-\beta} \\ &= \delta(1-\gamma)\left(\frac{\gamma\delta}{r}\right)^{\frac{\gamma}{1-\gamma}} \end{aligned}$$

*Appendix A.2. Classification of Dynamic Regimes*

This subsection provides the complete classification of possible dynamic regimes that can be observed according to Dynamic (5) and (6). Let us first highlight the following basic properties of Dynamic (5) and (6):

(1)   According to non-negativity constraint $E \geq 0$, state $E = 0$ is always a locally attractive stationary state in that, if stock $E > 0$ is low enough, then $\dot{E} < 0$ holds according to Equation (5);

(2)   State $E = \overline{E}$ is a stationary state if and only if $\overline{E} \geq 1/\Gamma$. That is, if carrying capacity $\overline{E}$ is higher than threshold value $1/\Gamma$ of $E$ that separates the regimes with and without specialization. Furthermore, no stationary state with $E > \overline{E}$ can exist because $\dot{E} < 0$ always holds for $E > \overline{E}$;

(3)   Interior stationary states (that is, those belonging to interval $(0, \overline{E})$) coincide with the values of $E$ that annul the right-hand side of Equation (5), in such stationary states, the economy is not specialized (that is, $1 > L^* > 0$);

(4)   According to Equation (5), $\dot{E} = 0$ holds if $f(E) = g(E)$, where:

$$f(E) := E(\overline{E} - E)$$

$$g(E) := \eta a\,(1 - \Gamma E)$$

Graphs of $f(E)$ and $g(E)$ can have at most two intersection points; therefore, at most, two interior stationary states exist. The complete taxonomy of possible dynamic regimes is illustrated in the following proposition. The proof is straightforward and therefore omitted.

**Proposition A1.**   *(1)   If condition $\eta < \eta_0 := 1/\left(a\Gamma^2\right)$ holds, then the following dynamic regimes can be observed:*

*(1a)   If $\overline{E} \geq \dfrac{1}{\Gamma}$, then a unique interior stationary state $E_A \in (0, 1/\Gamma)$ exists. This is repulsive and separates the basins of attraction of locally attractive stationary states $E = 0$ and $E = \overline{E}$ (see Figure 1b);*

(1b)  If $\frac{1}{\Gamma} > \overline{E} > \overline{E}_T := 2\sqrt{a\eta} - a\eta\Gamma$, then two interior stationary states $E_A$ and $E_B$ exist, with $0 < E_A < E_B < 1/\Gamma$. Repulsive stationary state $E_A$ separates the basins of attraction of locally attractive stationary states $E = 0$ and $E_B$ (see Figure 1b) (in the context of $\eta < \eta_0$, $\frac{1}{\Gamma} > \overline{E}_T > 0$ always holds);

(1c)  If $\overline{E} = \overline{E}_T$, then two stationary state exist, $E = 0$ and $E = E_T := \left(\overline{E} + \eta a\Gamma\right)/2 < 1/\Gamma$, and their basins of attraction are, respectively, intervals $[0, E_T)$ and $[E_T, +\infty)$;

(1d)  If $\overline{E} < \overline{E}_T$, then stationary state $E = 0$ is globally attractive (see Figure 1a).

(2)  If condition $\eta \geq \eta_0$ holds, the following dynamic regimes can be observed:

(2a)  If $\overline{E} > \frac{1}{\Gamma}$, then the dynamic regime coincides with that described in Point (1a) of this proposition (see Figure 2b);

(2b)  If $\overline{E} = \frac{1}{\Gamma}$, then two stationary states exist, $E = 0$ and $E = \overline{E}$, and their basins of attraction are, respectively, intervals $[0, \overline{E})$ and $[\overline{E}, +\infty)$;

(2c)  If $\overline{E} < \frac{1}{\Gamma}$, then stationary state $E = 0$ is globally attractive (see Figure 2a).

Coordinates of interior stationary states $E_A$ and $E_B$ (when existing) are given by:

$$E_A = \frac{\overline{E} + \eta a\Gamma}{2} - \frac{1}{2}\sqrt{(\overline{E} + \eta a\Gamma)^2 - 4\eta a}$$

$$E_B = \frac{\overline{E} + \eta a\Gamma}{2} + \frac{1}{2}\sqrt{(\overline{E} + \eta a\Gamma)^2 - 4\eta a} \tag{A8}$$

Note that in Subcase (1a) of the above proposition, if $\overline{E} = \frac{1}{\Gamma}$, then $E_B = \overline{E}$ holds. Furthermore, in Subcase (1c), $E_A = E_B = E_T$ holds; in such a case, straight line $f(E)$ is tangential to the graph of $g(E)$ at point $(E_T, g(E_T))$.

*Appendix A.3. Welfare of L-agents*

Bearing in mind that the equilibrium wage rate is $w = \delta(1 - \gamma)\left(\frac{\gamma\delta}{r}\right)^{\frac{\gamma}{1-\gamma}} = a(1 - \gamma)$ (see Appendix A.1), we can prove the following proposition.

**Proposition A2.** $\Pi_L(E) = E^\beta$ holds for $E \geq \frac{1}{\Gamma}$ while $\Pi_L(E) = \left[\Gamma^{-\beta} - a(1 - \gamma)\right]\Gamma E + a(1 - \gamma)$ holds for $E < \frac{1}{\Gamma}$, where $\left[\Gamma^{-\beta} - a(1 - \gamma)\right] > 0$ (see Equations (3) and (4)). That is, revenues $\Pi_L(E)$ of the representative L-agent are strictly increasing in E for every $E \geq 0$. Thus, in attractive stationary state $E = 0$, revenues are lower than in other attractive stationary states (either $E_B$ or $E = \overline{E}$) in the bistable regimes illustrated in Figures 1b,c and 2b.

**Proof.** Revenues $\Pi_L$ of the representative L-agent can be written as:

$$\Pi_L = E^\beta L^{*1-\beta} + (1 - L^*)w = E^\beta L^{*1-\beta} + a(1 - \gamma)(1 - L^*)$$

where $L^* = \min\{1, \Gamma E\}$. Thus $\Pi_L = E^\beta$ holds for $E \geq \frac{1}{\Gamma}$ while:

$$\begin{aligned}
\Pi_L &= E^\beta (\Gamma E)^{1-\beta} + a(1 - \gamma)(1 - \Gamma E) \\
&= \Gamma^{1-\beta}E - \Gamma E a(1 - \gamma) + a(1 - \gamma) \\
&= \left[\Gamma^{-\beta} - a(1 - \gamma)\right]\Gamma E + a(1 - \gamma)
\end{aligned}$$

holds for $E < \frac{1}{\Gamma}$. Furthermore, the condition:

$$\Gamma^{-\beta} - a(1-\gamma) = \frac{a(1-\gamma)}{1-\beta} - a(1-\gamma) > 0$$

is always satisfied. This implies that $\Pi_L$ strictly increases in $E$ for every $E \geq 0$. $\square$

In the following proposition, we focused on the trajectory starting from initial position $E(0) = \overline{E}$. The effects on revenues of L-agents generated by external mining investments can be better understood by comparing the dynamics generated by the model that is called Dynamics with the Mining Sector (DwMS), and the dynamics according to which $L = 1$ and $K_I = 0$ hold for every $E \geq 0$, which is called Natural Dynamics (ND). The latter is dynamics that would be observed in the absence of extractive operations financed by external investors. In the ND context, the environment is not affected by the polluting activity of mining, the time evolution of $E$ is described by logistic Equation (6), and the revenues of L-agents are given by function $\Pi_L^{ND}(E) := E^\beta$. Under this, $E = \overline{E}$ is always a globally attractive stationary state therefore, if the initial value $E(0)$ of $E$ coincides with $\overline{E}$, then $E = \overline{E}$ always holds.

In the DwMS context, we can see that if carrying capacity $\overline{E}$ is sufficiently low, and therefore dynamic regimes shown in Figures 1a and 2a occur (see Subcases (1d) and (2c) of Proposition A1 in Appendix A.2), then the trajectory starting from initial position $E(0) = \overline{E}$ converges to stationary state $E = 0$. In the remaining dynamic regimes, the trajectory starting from $E(0) = \overline{E}$ converges either to $E = \overline{E}$ (Figures 1c and 2b, and Subcases (1a), (2a), and (2b) of Proposition A1 in Appendix A.2) or to $E_B$ (Figure 1b, and Subcases (1b) and (1c) of Proposition A1). If the trajectory starting from $E(0) = \overline{E}$ approaches either $E_B$ or $E = 0$, then stock $E$ and revenues $\Pi_L(E)$ decrease along it, while mining external investments $K_I$ increase. However, this does not exclude the possibility that external mining investments may generate improvement in the welfare of the local population. The following proposition highlights conditions under which $\Pi_L^{ND}(\overline{E}) = \overline{E}^\beta$ is lower than the value of $\Pi_L(E)$, evaluated at stationary states $E = 0$ and $E_B$ of DwES.

**Proposition A3.** *In stationary state $E = 0$ of DwMS, workers' revenues are higher than in stationary state $E = \overline{E}$ of ND (that is, $\overline{E}^\beta < \Pi_L(0)$ holds) if and only if:*

$$\overline{E} < [a(1-\gamma)]^{\frac{1}{\beta}} = \left[ \delta(1-\gamma) \left( \frac{\gamma\delta}{r} \right)^{\frac{\gamma}{1-\gamma}} \right]^{\frac{1}{\beta}} \tag{A9}$$

*where $\left[ \delta(1-\gamma) \left( \frac{\gamma\delta}{r} \right)^{\frac{\gamma}{1-\gamma}} \right]^{\frac{1}{\beta}} < 1/\Gamma$, that is, if the carrying capacity of the environmental resource (measured by $\overline{E}$) is low enough. In stationary state $E_B$ of DwMS, workers' revenues are higher than in stationary state $E = \overline{E}$ of ND (that is, $\overline{E}^\beta < \Pi_L(E_B)$ holds) if and only if:*

$$E_B > \frac{\overline{E}^\beta - a(1-\gamma)}{[\Gamma^{-\beta} - a(1-\gamma)]\,\Gamma} \tag{A10}$$

*where $\Gamma^{-\beta} - a(1-\gamma) > 0$. Condition (A10) is satisfied if the environmental impact (measured by $\eta$) of the mining sector is low enough.*

**Proof.** The proof of this proposition is straightforward. We limited ourselves to proving the last statement. If the value of parameter $\eta$ decreases, then the value of $E_B$ increases therefore, Condition (A10) is more easily satisfied. It is thus possible to prove that, if the value of $\eta$ is low

enough, then Condition (A10) holds. This result can be checked by taking into account that Subcase (1b) of Proposition 4 is characterized by conditions:

$$1/\Gamma > \overline{E} > \overline{E}_T := 2\sqrt{a\eta} - a\eta\Gamma \quad \text{and} \quad \eta < \eta_0 := 1/\left(a\Gamma^2\right)$$

where values of $a$ and $\Gamma$ do not depend on $\eta$ while $\overline{E}_T \to 0$ for $\eta \to 0$. Thus, given $\overline{E} < 1/\Gamma$, such conditions are always satisfied if $\eta$ is low enough. Furthermore, if $\overline{E} < 1/\Gamma$, then $L < 1$ holds in $E = \overline{E}$; consequently, $\overline{E}^\beta < \Pi_L(\overline{E})$ (because, in the DwES context, revenues are maximized with respect to $L$ given the value of $E$). Thus, $\overline{E}^\beta < \Pi_L(E_B)$ holds if $E_B$ is near enough to $\overline{E}$ due to the continuity of function $\Pi_L(E)$. This is, indeed, precisely the case when $\eta$ is low enough, in that $E_B \to \overline{E}$ for $\eta \to 0$ (see Condition (A8)). $\quad\square$

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
