# Peer review of "Mining and Local Economies: Dilemma between Environmental Protection and Job Opportunities"

_sustainability, doi:10.3390/su11226244_

Round 1

Reviewer 1 Report

In my opinion, the presentation of graphs in Fig. 1-2 require descriptor of vertical axis, Fig. 4 - horizontal and vertical axis, Fig 5 - horizontal axis.

Author Response

Point 1: In my opinion, the presentation of graphs in Fig. 1-2 require descriptor of vertical axis, Fig. 4 - horizontal and vertical axis, Fig 5 - horizontal axis.

Response 1: done.

Reviewer 2 Report

Please clearly state that the article proposes a theoretical model or framework in the abstract and introduction. Please include any limitations of your model in the conclusion

Author Response

Point 1: Please clearly state that the article proposes a theoretical model or framework in the abstract and introduction. Please include any limitations of your model in the conclusion

Response 1: we added a sentence in the abstract to clarify that our contribution is a theoretical model. We further stressed and discussed our theoretical approach in the introduction.  In the last section, we included a paragraph which discusses the main limitations and how the model would be extended.  

Reviewer 3 Report

General comment

The topic is of interest. The social licence to operate (SLTO)  is a hotly discussed topic in the mining industry.The ‘social licence to operate’ (Prno 2013; Parsons and Moffat 2014) may become the most serious challenge for mining, smelting and refining in the future (acatec 2018b; Wellmer et al. 2018). In their annual risk radar of the top ten risk factors in the natural
resources industry, the consulting company EY classified the ‘Social Licence to Operate’ as the greatest risk for 2019 (EY 2019).

But SLTO is totally missing in this paper. The word does not appear at all. It is suggested to add this important aspect. A few references of a wide range of publications:

EY (2019) Business risks facing mining and metals 2018-2019.
https://assets.ey.com/content/dam/ey-sites/ey-com/global/topics/mining-metals/miningmetals-pdfs/ey-top-10-business-risks-facing-mining-and-metals-in-2019-20.pdf.

acatech (2018 a) National Academy of Science and Engineering, German National Academyof Sciences Leopoldina, Union of the German Academies of Sciences and Humanities Raw materials for the energy transition – Securing a reliable and sustainable supply. Position paper, Schriftenreihe Energiesysteme der Zukunft, Munich, Berlin (acatech), 100 p
https://www.acatech.de/Publikation/raw-materials-for-the-energy-transition-securing-areliable-and-sustainable-supply/.

Parsons R, Moffat K (2014) Constructing the Meaning of Social Licence. Social Epistomology 28/ 3–4: 340–363 

Prno J (2013) An Analysis of Factors leading to the Establishment of a Social Licence to Operate in the Mining Industry. Resources Policy 38: 577–590

Wellmer FW, Buchholz P, Gutzmer J, Hagelüken C, Herzig P, Littke R, Thauer RK (2018) Raw Materials for Future Energy Supply. Springer, Berlin, Heidelberg, 255

See also a very recent article in the Bulletin of the AusIMM (August 2019): Jeyaretnam, T: Social aspects of mine closure.

Detailed comments

1.) The quotion and the list of references is not correct: The guidelines for authors clearly states "Authors must be listed in the same order as they appear in the original document.". The authors did not do this. They have the list of authors in the references done according to the ABC and then quoted numbers in the text which now of course are very irregular.

2.) For internet and www-quotes generally there are no access date given.

2.) Line 17, the abbreviation TNC and BRIC has to be explained and the countries given which belong to BRIC (is South Africa included or not?) .

3.) Line line 17: The statement "The main mining companies are TNCs from advanced and newly-rich BRIC countries can easily be misunderstood. What you probably mean is "from newly-rich BRIC countries or advanced industrialized countries.". 6 out of the 10 largest mining companies --and 31 out of 50-- do not belong to BRIC countries. (https://www.mining.com/top-50-biggest-mining-companies/)

4.) line 20 not company but companies (plural)

5.) Line 20: Public Eye is in Capital Letters. But the reference [34] is not Public Eye.

6.) Line 23/24 quotation and access date should be  in the references

7.) Line 26 Abbreviation ICMM: International Council on Mining & Metals is given in line 16. Give already there in brackets the abbreviation.

8.) Line 57, Footnote 1. Are there really no more up-to date figure than these 10 years old ones. I cannot believe it.

9.) Line 63: Why only external companies, see also your footnote 2.

10.) Chapter 2: The model setup.  You have to be aware, that you are writing to for a general readership interested in sustainability problems. You are not writing for a readership which is specialized in economic models. You have to explain this far better:

For example line 97 b: "The production function of the two sectors are concave, increasing and homogeneous of degree 1 in their inputs". Why ? This has to be explained far better.

or: line 104: Cobb-Douglas function: What is it?

Author Response

Point 1: The topic is of interest. The social licence to operate (SLTO)  is a hotly discussed topic in the mining industry.The ‘social licence to operate’ (Prno 2013; Parsons and Moffat 2014) may become the most serious challenge for mining, smelting and refining in the future (acatec 2018b; Wellmer et al. 2018). In their annual risk radar of the top ten risk factors in the natural
resources industry, the consulting company EY classified the ‘Social Licence to Operate’ as the greatest risk for 2019 (EY 2019).

But SLTO is totally missing in this paper. The word does not appear at all. It is suggested to add this important aspect. A few references of a wide range of publications:

EY (2019) Business risks facing mining and metals 2018-2019.
https://assets.ey.com/content/dam/ey-sites/ey-com/global/topics/mining-metals/miningmetals-pdfs/ey-top-10-business-risks-facing-mining-and-metals-in-2019-20.pdf.

acatech (2018 a) National Academy of Science and Engineering, German National Academyof Sciences Leopoldina, Union of the German Academies of Sciences and Humanities Raw materials for the energy transition – Securing a reliable and sustainable supply. Position paper, Schriftenreihe Energiesysteme der Zukunft, Munich, Berlin (acatech), 100 p
https://www.acatech.de/Publikation/raw-materials-for-the-energy-transition-securing-areliable-and-sustainable-supply/.

Parsons R, Moffat K (2014) Constructing the Meaning of Social Licence. Social Epistomology 28/ 3–4: 340–363 

Prno J (2013) An Analysis of Factors leading to the Establishment of a Social Licence to Operate in the Mining Industry. Resources Policy 38: 577–590

Wellmer FW, Buchholz P, Gutzmer J, Hagelüken C, Herzig P, Littke R, Thauer RK (2018) Raw Materials for Future Energy Supply. Springer, Berlin, Heidelberg, 255

 See also a very recent article in the Bulletin of the AusIMM (August 2019): Jeyaretnam, T: Social aspects of mine closure.

Response 1: we thank the referee for this suggestion and these indications of references. We have referred to the topic of “social licence to operate” in the introduction in order to highlight the relevance of the interactions with local populations and activities from the perspective of mining companies.  We added the references to Prno J (2013), Parsons and Moffat (2014), Andrews et al. (2017), Acatech (2018), EY (2019). 

Detailed comments

Point 1: The quotion and the list of references is not correct: The guidelines for authors clearly states "Authors must be listed in the same order as they appear in the original document.". The authors did not do this. They have the list of authors in the references done according to the ABC and then quoted numbers in the text which now of course are very irregular.

Response 1: thank you, we have corrected the order of the references.

Point 2: For internet and www-quotes generally there are no access date given.

Response 2: we added the date of access.

Point 3: Line 17, the abbreviation TNC and BRIC has to be explained and the countries given which belong to BRIC (is South Africa included or not?) .

Response 3: we have substantially revised the introduction and we cut both acronyms

Point 4: Line line 17: The statement "The main mining companies are TNCs from advanced and newly-rich BRIC countries can easily be misunderstood. What you probably mean is "from newly-rich BRIC countries or advanced industrialized countries.". 6 out of the 10 largest mining companies --and 31 out of 50-- do not belong to BRIC countries. (https://www.mining.com/top-50-biggest-mining-companies/)

Response 4: we have decided to cut this sentence since it was not strictly necessary and we have introduced additional sections in the paper in order to take into account suggestions from other reviewers.

Point 5: line 20 not company but companies (plural):

Response 5: we have corrected the error. Thank you.

Point 6: Line 20: Public Eye is in Capital Letters. But the reference [34] is not Public Eye.

Response 6: we cut the sentence.

Point 7: Line 23/24 quotation and access date should be in the references

Response 7: done.

Point 8: Line 26 Abbreviation ICMM: International Council on Mining & Metals is given in line 16. Give already there in brackets the abbreviation.

Response 8: done.

Point 9: Line 57, Footnote 1. Are there really no more up-to date figure than these 10 years old ones. I cannot believe it.

Response  9: we have updated information on agricultural employment. Now the figure refers to 2018.

Point 10: Line 63: Why only external companies, see also your footnote 2.

Response 10: We cut the world “external” in line 63.

Point 11: Chapter 2: The model setup.  You have to be aware, that you are writing to for a general readership interested in sustainability problems. You are not writing for a readership which is specialized in economic models. You have to explain this far better:

For example line 97 b: "The production function of the two sectors are concave, increasing and homogeneous of degree 1 in their inputs". Why ? This has to be explained far better.

or: line 104: Cobb-Douglas function: What is it?

Response 11: we have revised the first paragraph of section “The model setup”. We extended the explanation of the local sector and capital costs for investors.  We explained that the production functions are increasing at decreasing rate in their input and we replaced the term “Cobb-Douglas” function with an expression that describes its main characteristic, namely the fact that it models a production technique with constant returns to scale.  We gave more details about the logistic function used to model renewable resources and we added a reference for additional explanations the reader may want.

We explicitly describe the differences between renewable natural resources which are production factors for the local sector, and natural (mineral) resources used by the mining sector. In particular, since we explain that we concentrate on the medium-term time horizon and assume that mining does not experience increasing costs due to mineral resource exhaustion, the role of mineral resources is implicitly considered in the parameter δ. We added two sentences to clarify this point.

In order to be of interest to a non-technical audience, throughout the paper, we cut several sentences, which provided technicalities and analytical clarifications with limited implications in terms of economic meaning. For instance, we removed the sentence on regimes that are not robust, in that they occur if equality conditions in terms of parameter values are satisfied. These cases are residual and were introduced in the analysis only for the sake of completeness. In addition, we moved a part of the analysis to the Appendix. Now the main text concentrates on the motivation and presentation of the model, the main analytical results and their comments. Propositions 2 and 3, for instance, have been moved into the Appendix.

Reviewer 4 Report

I suggest that you dedicate a separate section to discuss the significance of your findings and policy implications of your findings, as well as linking your findings to the existing literature. 

Author Response

Point 1: I suggest that you dedicate a separate section to discuss the significance of your findings and policy implications of your findings, as well as linking your findings to the existing literature. 

Response 1: We addressed this comment by making a substantial revision of the first part of the paper and organizing the text in two sections, the introduction and the literature review. The first one presents our motivation, the background and the research gaps that we intend to address. The second one discusses the related literature showing in what aspects our model can give contribution to the debate on the impact of mining activities on local rural populations.

In addition, in the section that discusses the results of the model analysis, we clarified the differences between the three different dynamic regimes explaining their conditions of existence, but with a specific focus on their local welfare outcomes. We explained in detail their features and we used concrete examples of mining communities to exemplify their differences.

We rewrote and extended the conclusions. We suggested that future empirical research should investigate, more systematically than in the past, the economic role of mining’s environmental pressure on rural economic activities in developing countries. In addition, we clarified the different risks that rural local economies could face when there is openness towards large-scale mines in case of the low, high and intermediated levels of carrying capacity, that is, respectively, the risks of dependence on exogenous forces, local opposition and welfare-reducing specialization in the mining sector. Finally, we have specified that, in the context of intermediated environmental carrying capacity, policies for pollution control is a sine qua non for welfare gains, despite new job opportunities and no barriers to employment.

Reviewer 5 Report

The paper appears to be a theoretical contribution to the analysis of welfare impacts of mining activities. Research of this type, especially where models are concerned, should be based on a vast account of literature and links to existing theories which is clearly missing. It is therefore difficult to assess the novelty and the value added of the manuscript. This situation needs to be rectified if the paper is to be published.

Please, incorporate the answers to following questions with the necessary citations:

1) What economic models are usually used to describe / assess welfare impacts and environmental damage? What is the main mainstream one / type of model?

2) How different is your model from it? What new features does your model contain? What assumptions are added / disregarded?

3) (optional) Can it be confronted with data / tested? If yes, to what degree is it robust?

Furthermore, I recommend to add a discussion section into the paper where a comparison with previous approaches, at least the ones mentioned in the introduction is performed.

Author Response

Point 1:The paper appears to be a theoretical contribution to the analysis of welfare impacts of mining activities. Research of this type, especially where models are concerned, should be based on a vast account of literature and links to existing theories which is clearly missing. It is therefore difficult to assess the novelty and the value added of the manuscript. This situation needs to be rectified if the paper is to be published.

Please, incorporate the answers to following questions with the necessary citations:

1) What economic models are usually used to describe / assess welfare impacts and environmental damage? What is the main mainstream one / type of model?

2) How different is your model from it? What new features does your model contain? What assumptions are added / disregarded?

3) (optional) Can it be confronted with data / tested? If yes, to what degree is it robust?

Furthermore, I recommend to add a discussion section into the paper where a comparison with previous approaches, at least the ones mentioned in the introduction is performed.

Response 1: We addressed this comment by making a substantial revision of the first part of the paper and organizing the text in two sections, the introduction and the literature review. The first one presents our motivation, the background and the research gaps that we intend to address. The second one discusses the related literature showing in what aspects our model can give a contribution to the debate on the impact of mining activities on local rural populations. Since the literature review is in a separate section, we extended it in order to compare more systematically our contribution to both empirical and theoretical literature. Specifically, we identified three type of channels of interaction between large mining companies and local populations: the market channel, the fiscal channel and the environmental channel. We have focused on the main mechanisms involved in the first and second one (employment generation and environmental externalities on local activities, respectively) . We have explained our choice by referring to recent literature reviews. We have highlighted that, to the best of our knowledge, the idea of analysing the economic scope of the environmental channel and of jointly considering the environmental and market dynamics is an element of novelty.

We have also divided the explanation of analytical results from the final remarks. We believe that the model results cannot be directly confronted with empirical data, but in the section that discusses the results of the model analysis, we use concrete cases of mining communities to exemplify the differences between the three dynamic regimes.

This is the list of additional references cited in the literature review and in the section with the comments on the analytical results:

Acatech -- National Academy of Science and Engineering, German National Academy of Sciences Leopoldina, and Union of the German Academies of Sciences and Humanities. 2018. Raw materials for the energy transition -- Securing a reliable and sustainable supply. Position Paper of the Academies' Project "Energy Systems of the Future", available at https://energiesysteme-zukunft.de/en/position-paper/raw-materials-for-the-energy-transition/, accessed in October 2019.

Acemoglu, D. 2008. Introduction to Modern Economic Growth, Princeton University Press, Princeton

Aitken, D., Rovera, D., Godoy-Fa_undez, A., Holzapfel, E. 2016. Water Scarcity and the Impact of the Mining and Agricultural Sectors in Chile. Sustainability 8(128).

Akabzaa, T. 2009. Mining in Ghana: implications for national economic development and poverty reduction. In: Campbell, B. (Ed.), Mining in Africa: Regulation and Development. IDRC, Canada, pp. 25-65

Álvarez, R., García Marín, Á.G. and Ilabaca, S. (in press. Available online 10 May 2018) Commodity price shocks and poverty reduction in Chile. Resources Policy.

Andrews, T., Elizalde, B., Le Billon, P. Hoon Oh, C., Reyes, D. and I. Thomson. 2017. The Rise in Conflict Associated with Mining Operations: What Lies Beneath? Canadian International Resources and Development Institute.

Andrews, N. 2018. Land versus livelihoods: Community perspectives on dispossession and marginalization in Ghana's mining sector, Resources Policy 58: 240{249.

Arias, M., Atienza, M. and Cademartori, J. 2014 Large mining enterprises and regional development in Chile: between the enclave and cluster. Journal of Economic Geography 14, 73—95

Aroca, P. 2001. Impacts and development in local economies based on mining: The case of the Chilean II region, Resources Policy 27, 119-134.

Ayee, J., Soreide, T., Shukla, G.P. and Le, T.M., 2011. Political Economy of the Mining Sector in Ghana. World Bank Policy Research Working Paper No. 573.

Bravo-Ortega, C. and Munoz, L. (In press, available online 12 June 2018). Mining services suppliers in Chile: A regional approach (or lack of it) for their development. Resources Policy.

Clark, C. 1990. Mathematical Economics, New York: JohnWiley and Sons.

Cuba, N., Bebbington, A., Rogan, J. and Millones, M. 2014. Extractive industries, livelihoods and natural resource competition: Mapping overlapping claims in Peru and Ghana, Applied Geography 54, 250-261.

Der Goltz, J.V. and Barnwal, P. 2019. The local wealth and health effects of mineral mining in developing countries. Journal of Development Economics, 139(C), 1-16.

Di Corato L., 2013. Profit sharing under the threat of nationalization. Resources and Energy Economics 35, 295–315.

Duraiappah, A.K. 1998. Poverty and environmental degradation: A review and analysis of the nexus. World Development, 26(12): 2169-2179.

2019. Top 10 business risks facing mining and metals in 2019-20. Available at https://www.ey.com/en_gl/mining-metals/10-business-risks-facing-mining-and-metals, accessed in October 2019.

Ghebrihiwet N., Motchenkova E., 2017. Relationship between FDI, foreign ownership restrictions, and technology transfer in the resources sector: a derivation approach. Resources Policy 52, 320–326.

Hausermann, H., Ferring, D., Atosona, B., Mentz, G., Amankwah, R., Chang, A., Hartfield, K., Effah, E., Asuamah, G.Y., Mansell, C. and Sastri, N. 2018. Land-grabbing, land-use transformation and social differentiation: Deconstructing "small-scale" in Ghana's recent gold rush. World Development 108, 103—114

Karakaya, E. and Nuur, C. 2018. Social sciences and the mining sector: Some insights into recent research trends. Resources Policy 58, 257-267.

Katz, J. and Pietrobelli, C. 2018. Natural resource based growth, global value chains and domestic capabilities in the mining industry. Resources Policy 58, 11-20.

Issabayev M., Pelzman J., 2019. A model of FDI spillover in a natural resource rich LDC. Resources Policy 64, 101479.

Issabayev M., Rizvanoghlu I., 2019. Optimal choice between local content requirement and fiscal policy in extractive industries: a theoretical analysis. Resources Policy 60, 1-8.

Lagos, G. 1997. Developing national mining policies in Chile: 1974-96. Resources Policy 23, No. 1/2, 51-69.

Lagos, G. and Blanco, E. 2010. Mining and development in the region of Antofagasta. Resources Policy 35, 265-275.

Macatangay R.E., 2016. Optimal local content requirement policies for extractive industries. Resources Policy 50, 244–252.

Mancini, L. and Sala, S. 2018. Social impact assessment in the mining sector: Review and comparison of indicators frameworks. Resources Policy 57, 98-111.

McMahon, G. and Remy, F. 2001. Large Mines and the Community: Socioeconomic and Environmental Effects in Latin America, Canada and Spain. Washington, DC: World Bank and the International Development Research Centre.

Mkodzongi, G. and Spiegel, S. 2019. Artisanal Gold Mining and Farming: Livelihood Linkages and Labour Dynamics after Land Reforms in Zimbabwe, The Journal of Development Studies, 55:10, 2145-216.

Mishra, P.P. and Pujari, A.K. 2008. Impact of mining on agricultural productivity: A case study of the Indian State of Orissa. South Asia Economic Journal 9(2), 337-350.

Ozkaynak, B., Rodriguez-Labajos, B., Arsel, M., Avc_, D., Carbonell, M.H., Chareyron, B., Chicaiza, G., Conde, M., Demaria, F., Finamore, R., Kohrs, B., Krishna, V.V., Mahongnao, M., Raeva, D., Singh, A.A., Slavov, T., Tkalec, T., Y_anez, I., Walter, M. and _Ziv_ci_c, L., 2012. Mining Conficts around the World: Common Grounds from Environmental Justice Perspective, EJOLT Report No. 7.

Parsons, R. and Moffat, K. 2014. Constructing the Meaning of Social Licence. Social Epistemology, 28(3-4), 340-363.

Prno, J. 2013. An analysis of factors leading to the establishment of a social licence to operate in the mining industry, ResourcesPolicy 38, 577--590.

Queirolo, F., Stegen, S., Restovic, M., Paz, M., Ostapczuk, P., Schwuger, M.J. and Munoz, L. 2000. Total arsenic, lead, and cadmium levels in vegetables cultivated at the Andean villages of northern Chile. The Science of the Total Environment 255, 75-84.

Ravi, K. J., Cui, Z., and Domen, J. K. 2016. Environmental Impact of Mining and Mineral Processing. Management, Monitoring, and Auditing Strategies. Elsevier Inc. Oxford, 2016.

Reichl, C., Schatz, M. and Zsak, M. World Mining Data, Vol. 32, Minerals Production, International Organizing Committee for World Mining Congresses, Vienna, 2017.

Romero, H., Mendez, M. and Smith, P. 2012. Mining Development and Environmental Injustice in the Atacama Desert of Northern Chile. Environmental Justice 5(2), 70-76.

Schueler, V., Kuemmerle, T. and Schroeder, H. 2011. Impacts of surface gold mining on land use systems in Western Ghana. Ambio 40, 528-539.

Scialabba, N. EH. and Hattam, C. (Eds) 2002. Organic agriculture, environment and food security. Food & Agriculture Org. 2002.

Temper, L., del Bene, D. and Martinez-Alier, J. 2015. Mapping the frontiers and front lines of global environmental justice: the EJAtlas. Journal of Political Ecology 22: 255-278.

Tuokuu, F.X.D, Gruber, J.S., Idemudi, U. and Kayir, J. 2018. Challenges and opportunities of environmental policy implementation: Empirical evidence from Ghana's gold mining sector, Resources Policy, 59, 435-445.

Round 2

Reviewer 3 Report

Line 52: I would not say "mining" is dirty. I suggest "perceived as dirty"

Line 114 "Nash barganing theory"  Explain!

Line 117 FDI. Explain abbreviation.

Author Response

I would not say "mining" is dirty. I suggest "perceived as dirty"

we changed the text following the reviewer's suggestion 

"Nash bargaining theory" Explain!

Please, see footnote 3.

FDI. Explain abbreviation.

We replaced "FDI policies" with "foreign direct investment policies". 

Reviewer 5 Report

My comments were incorporated.

Author Response

Thank you.